# NF-κB in Cancer Immunity: Friend or Foe?

**DOI:** 10.3390/cells10020355

**Published:** 2021-02-09

**Authors:** Guilhem Lalle, Julie Twardowski, Yenkel Grinberg-Bleyer

**Affiliations:** Cancer Research Center of Lyon, UMR INSERM 1052, CNRS 5286, Université Claude Bernard Lyon 1, Centre Léon Bérard, 69008 Lyon, France; guilhem.lalle@lyon.unicancer.fr (G.L.); julie.twardowski@lyon.unicancer.fr (J.T.)

**Keywords:** onco-immunology, immunotherapy, NF-kappaB

## Abstract

The emergence of immunotherapies has definitely proven the tight relationship between malignant and immune cells, its impact on cancer outcome and its therapeutic potential. In this context, it is undoubtedly critical to decipher the transcriptional regulation of these complex interactions. Following early observations demonstrating the roles of NF-κB in cancer initiation and progression, a series of studies converge to establish NF-κB as a master regulator of immune responses to cancer. Importantly, NF-κB is a family of transcriptional activators and repressors that can act at different stages of cancer immunity. In this review, we provide an overview of the selective cell-intrinsic contributions of NF-κB to the distinct cell types that compose the tumor immune environment. We also propose a new view of NF-κB targeting drugs as a new class of immunotherapies for cancer.

## 1. Introduction: The Growing Role of Immunity in Cancer

The past decades have been marked by a remarkable interest in research on onco-immunology, i.e., the interactions between cancer and the immune system. This has culminated with the approval of anti-Cytotoxic T-Lymphocyte-Associated protein 4 (CTLA-4) and Programmed Death (PD)-1/PD-Ligand 1 monoclonal antibodies (mAbs) and CD19-targeted Chimeric Antigen Receptor (CAR)-T-cell infusions as frontline therapies for a variety of cancers. Despite the fact that these treatments are not effective in all patients and all types of diseases, they established the revolutionary concept that the immune system can be manipulated to treat cancer. Many different immune cell types, surface receptors and secreted factors are involved in the different steps of cancer progression. In order to highlight novel efficient and safe immunotherapies, it is necessary to apprehend this complexity of immune responses to cancer. Recent technological developments such as single-cell sequencing, high-dimensional flow and mass cytometry, and multiparameter immunofluorescence techniques, have enabled the fine dissection of immune cell subsets, their phenotype and function, revealing a landscape even more complicated than originally thought. However, little is known about the intracellular pathways that dictate the development and fate of different cell subsets in the tumor environment. It is becoming evident that understanding these intracellular cascades and transcription factors may highlight potential therapeutic targets. Among those, the Nuclear Factor kappa light chain enhancer of activated B cells (NF-κB) transcription factors exhibit various and sometimes confusing effects on most immune subsets involved in immune responses to cancer.

## 2. A Simplified View of the Signaling to NF-κB

NF-κB is a family of 5 transcription factors that all share a Rel Homology Domain (RHD): NF-κB1 (p105/p50, encoded by *Nfkb1*), NF-κB2 (p100/p52, *Nfkb2*), RelA (*Rela*), RelB (*Relb*) and c-Rel (*Rel*) (reviewed in [1]). In the resting state, NF-κB subunits are retained in the cytosol by IκB proteins and by unprocessed p105 and p100 that act as inhibitors through the ankyrin repeats in their C-terminal portion. The engagement of diverse receptors leads to the nuclear translocation of homo- or heterodimers of NF-κB subunits, which can then activate or repress gene transcription.

The signaling cascade leading to NF-κB activation was historically split in two pathways. In innate immune cells, activation of the canonical pathway is classically driven by the engagement of Pattern Recognition Receptors (PRRs) such as Toll-Like Receptors (TLRs). This leads to the recruitment of adaptors such as MyD88 or Toll/IL-1R domain-containing adaptor-inducing IFN-β (TRIF) and the activation of Tumor necrosis factor Receptor–associated Factor (TRAF)6. Similarly, cytokine receptors or members of the Tumor Necrosis Factor Receptor superfamily (TNFRSFs) can engage TRAF2/5. Additionally, in adaptive immune cells, engagement of the antigen receptor (BCR and TCR for B and T cells, respectively) recruits Protein Kinase C (PKC)β/PKCθ to enable the formation of the Caspase Recruitment domain-containing Membrane-Associated guanylate kinase protein-1 (Carma1)/ B-cell lymphoma/leukemia 10 (Bcl10) and Mucosa-Associated Lymphoid Tissue lymphoma translocation protein 1 (Malt1), or CBM complex [2]. All these signals lead to the activation of the IκB kinase (IKK) complex (composed of the α and β catalytic subunits and γ scaffolding protein), which by phosphorylating Inhibitor of NF-κB (IκBα) and p105 triggers their proteasomal degradation and processing to the active p50 protein, respectively. This culminates in the rapid and transient translocation of NF-κB subunits to the nucleus. The alternative NF-κB pathway is activated in response to the engagement of a number of TNFRSFs, such as CD40 on dendritic cells (DCs) and B cells, B cell-Activating Factor Receptor (BAFF-R) on B cells or TNFR2 on T cells. Through the recruitment of TRAF2/3, this allows the stabilization of NF-κB-inducing kinase (NIK) and the activation of IKKα/α homodimers that phosphorylate p100 and lead to its processing into active p52. p52/RelB dimers are then translocated to the nucleus. Importantly, converging evidence now indicates that the canonical and alternative pathways, as well as subunits, are highly intricate. Hence, it may be more relevant to report the composition and roles of NF-κB subunits and dimers, rather than oversimplifying NF-κB to two distinct pathways.

As one of the most studied family of transcription factors, NF-κB has been highlighted for its diverse roles in most biological processes, with a particular focus on inflammation, adaptive immunity and oncogenesis. Indeed, extensive research has identified both cell-autonomous and cell-extrinsic roles for NF-κB in cancer. For instance, canonical NF-κB controls tumor cell proliferation, tumor-mediated angiogenesis as well as metastasis. This is nicely demonstrated by the reduced tumor burden detected in mouse models of K-Ras induced lung cancer carrying conditional ablation of *Ikbkb* or *Rela*, or transgenic expression of an IκBα-superrepressor; similar observations were made in melanoma among other cancers [3,4,5,6]. However, opposite results were described in murine liver cancer or squamous cell carcinoma, where canonical NF-κB inhibition increased tumor growth, thus highlighting the cell type-specific functions of this pathway and a possible role for NF-κB as a tumor suppressor in certain settings [7,8]. In addition to these cell-intrinsic functions, NF-κB activation in tumors appears to impact their immune environment. It is well established that tumor cells produce a number of NF-κB-dependent cytokines and chemokines that affect the recruitment and phenotype of immune cells in the tumor bed, and the outcome of cancer [9,10,11,12]. Moreover, NF-κB controls the mRNA expression and protein stability of PD-L1 in tumor cells, thereby promoting the inhibition of cytotoxic CD8^+^ T cells [13,14]. Finally, NF-κB plays many cell-autonomous roles in the function of immune cells in the tumor environment, clearly impacting cancer outcome. This will be the focus of the present review.

## 3. NF-κB in Innate Immunity and Inflammation

The development and progression of some tumors, such as colon or liver cancers, have been tightly associated with inflammatory cells and mediators. Because NF-κB, through its function in innate immune cells, is a critical orchestrator of the inflammatory process, it is often viewed as the main molecular link between inflammation and carcinogenesis. However, its roles may be multifaceted depending on the cell subset, the type of cancer and the signaling pathways that are engaged (Figure 1).

### 3.1. Pro-Tumoral Roles of NF-κB in Innate Immune Cells: Macrophages and MDSCs

#### 3.1.1. Macrophages

Tumor-Associated Macrophages (TAMs) are a major component of the immune infiltrate of solid tumors [15]. Different flavors of macrophages can be found and often co-exist in the tumor microenvironment (TME): ‘M1′ macrophages are potent producers of pro-inflammatory factors, whereas ‘M2′ macrophages display anti-inflammatory, immunosuppressive and pro-angiogenic properties [16]. Ordinarily, high TAM infiltration is correlated with bad prognosis in many cancers irrespective of their polarization status [17].

In vitro assays have unequivocally demonstrated the role of canonical NF-κB in the expression of inflammatory cytokines by macrophages- in other words, polarization towards an M1 phenotype [16,18]. Engagement of the TLR-Myd88 axis, as well as TNFR1, enables transcription of *Tnf*, *Il6* or *Il12b* among others, through canonical NF-κB. In contrast, p50 homodimers have regulatory activity, which is essential for an M2 “tolerant” phenotype in vitro and in vivo [19,20]. Alternative NF-κB subunits seem to be expendable for the expression of inflammatory cytokines [21]; instead overexpression of RelB inhibits *Tnf* expression in M1-polarizing conditions [22].

Seminal studies from Michael Karin’s group have established the pro-tumoral function of the canonical NF-κB pathway in macrophages, in the context of inflammation-driven cancer. Ablation of *Ikbkb* or *Rela* in macrophages using the Lysozyme (LysM)-CRE strain, leads to decreased tumor growth in mouse models of cigarette-smoke induced lung cancer and colitis-associated colon cancer [23,24,25]. Conversely, in models of breast, ovarian cancer and glioblastoma, canonical NF-κB activation in macrophages has an anti-inflammatory effect, preventing antitumor immunity and allowing tumor escape, in particular through the secretion of interleukin (IL)-10 [26,27,28,29]. Thus, depending on the tumoral setting, NF-κB exerts its deleterious functions by promoting either inflammation or immunosuppression; these separate functions have been discussed in the past [30]. Of note, this pathogenic role of canonical NF-κB was confirmed in patients with lung cancer, in which RelA activation in lung myeloid cells is associated with poor survival [25]. At odds with these reports, it was proposed that IKKβ may exert a protective function against tumor cell dissemination. Indeed, metastasis of breast cancer and melanoma cells was drastically increased in mice with myeloid-restricted deletion of *Ikbkb*, due to impaired innate and adaptive immunity to tumors [31,32,33]. Nevertheless, opposite results were found using a similar model of breast cancer metastasis [29]. These discrepancies may be attributable to the generalized use of the LysM-CRE strain in macrophage biology, which drives expression of the recombinase not only in macrophages but also in highly immunosuppressive MDSCs that are described below [34]. These complex functions of NF-κB could also be harnessed in a therapeutic perspective. Cell therapy using *Ikbkb* or *Nfkb1*-deficient macrophages favors strong anti-tumor immunity and tumor rejection [27,35]. Conversely, as discussed in later sections, in vivo stimulation of TLRs using agonistic molecules, has a strong therapeutic potential in cancer, perhaps through its effect on macrophages [36]. Finally, the natural flavonoid baicalin, promotes the repolarization of TAM to M1 macrophages by increasing Relb and p52 expression, leading to decreased growth of hepatocellular carcinoma in mice [37].

Taken together, further exploration is needed to fully establish the role of NF-κB in the different tumor types and steps of cancer development. Deciphering these precise molecular mechanisms, in particular the selective roles of NF-κB subunits, will help provide a rationale for the targeting of this pathway in cancer. In the future, the development of more specific tools (conditional gene ablation, cell depletion, etc.) will undoubtedly clarify these dual functions.

#### 3.1.2. MDSCs

Myeloid-Derived Suppressor Cells (MDSCs) have emerged in the past years as an important player in cancer immunity. They compose a heterogeneous subset of innate immune cells defined by their immunosuppressive properties, in particular their expression of IL-10, Transforming Growth Factor (TGF)-β, or Indolamine 2,3 DiOxygenase (IDO) among other factors, rather than by their phenotype that can overlap with other myeloid subsets [38]. Consequently, high MDSC infiltration is usually correlated with a poor prognostic value in cancer [39]. In mice, depletion of MDSCs (using poorly specific anti-Gr1 or anti-Ly6C mAbs) leads to enhanced antitumor immunity and reduced tumor burden [40,41]. The biology of MDSCs and the therapeutic potential of their targeting is currently under scrutiny, NF-κB appearing to largely support their deleterious function.

It was proposed that NF-κB could support the activation and expansion of MDSCs [42,43]. In pre-clinical models of cancer, both the canonical and the alternative pathway were reported to be implicated. It was first shown that TLR2/TLR4- MyD88-mediated signaling was required for the optimal function of MDSCs and suppression of anti-tumor immunity in models of colon cancer and fibrosarcoma, suggesting an involvement of canonical NF-κB activation [44,45]. Similarly, TNFR/TNFR2 signaling promotes MDSC function, especially IL-10 production, in an NF-κB-dependent manner [46]. More directly, the inhibitory function of MDSCs was strongly reduced in *Nfkb1*^−/−^ animals- though their number was increased in the tumor bed- resulting in the reduced growth of transplanted murine fibrosarcoma [47]. In addition, LysM-CRExRela-Flox mice transplanted with glioblastoma cells showed reduced levels of MDSCs and pro-tumoral TAMs, suggesting a role for RelA in MDSC recruitment [28]. Finally, a recent study elegantly described the role of c-Rel in regulating the development, function and metabolism of MDSCs during tumorigenesis [48]. Indeed, LysM-CRExRel-Flox mice display reduced growth of melanoma and thymoma cells, and the administration of the specific c-Rel inhibitor R96A delayed tumor growth and potentiated anti-PD1 therapy. This fully established the critical function of all three canonical NF-κB subunits in the deleterious functions of MDSCs. Moreover, activation of the NIK/RelB-p52 axis induces *IDO* expression in human MDSC-like cells cultured with breast cancer cells, suggesting that alternative NF-κB subunits may also promote immunosuppression in tumors through its role in MDSCs [49].

Of note, some literature also supports a pro-tumoral role for NF-κB in neutrophils during cancer. The high activation of the canonical pathway in Tumor-Associated Neutrophils (TAN) suggests its pro-tumoral role by inhibiting apoptosis of TANs, promoting autophagy and inducing a pro-inflammatory state [50,51,52,53]. Finally, indirect evidence highlights a role for NF-κB in Neutrophil extracellular traps (NET) production [51], a known propagator of metastasis [54]. Altogether, this large body of evidence demonstrates the deleterious facet of NF-κB activation in innate immune cells through its effects on macrophages and MDSCs. However, the picture is way more complex than it seems and NF-κB can also suppress tumor growth through its pivotal functions in other innate immune subsets: DCs and NK cells.

### 3.2. NF-κB in the Promotion of Anti-Tumor Immunity: DCs and NK Cells

#### 3.2.1. DCs

DCs are the main bridge between innate and adaptive immune responses. Through their critical function as Antigen Presenting Cells (APCs) and their cytokine and chemokine expression, they can shift the balance between tumor immunity and tolerance [55]. DCs can come in different flavors defined by their phenotype and function in the TME. Type 1 Conventional DCs (cDC1s) have an optimal ability to cross-present tumor antigens to CD8^+^ T cells and produce high amounts of IL-12, making them potent anti-tumor cells [56], and the accumulation of cDC1s in the TME is of good prognosis in many cancers [57]. Furthermore, cDC1s are required for the response to anti PD-1 therapies in mice [58]. Type 2 cDCs (cDC2s) are classically associated with the priming of CD4^+^ T cells. Interestingly, intra-tumoral cDC2 density was also shown to be correlated with abundant CD4^+^ T cells and anti-PD-1 therapy response in melanoma patients [59]. Finally, plasmacytoid DCs (pDCs), characterized by their high expression of type I interferons, have complex roles in tumor immunity that largely rely on the type of stimulation and their activation status [60,61].

Extensive research has highlighted the various roles of NF-κB in the biology of DCs. This stems from the fact that most of the stimuli that induce DC maturation (PRR, cytokines, etc.) are known activators of the NF-κB pathways (reviewed in [62]). Briefly, canonical NF-κB subunits were historically described as important regulators of DC maturation and survival following TLR and CD40 stimulation [63,64,65], though this is still controversial and may in fact depend on the type of stimuli and DC subset [66]. In contrast, it is established that c-Rel is absolutely critical for the expression of *Il12p35* (which, together with *Il12p40*, encodes IL-12), which emphasizes its role in the function of cDC1s [63,66,67]. The NIK/RelB-NF-κB2 axis is also involved in DC biology, particularly for the stability of the cDC2 lineage, and is required for optimal antigen cross-presentation [68,69,70,71]. To add another layer of complexity, it was shown that canonical NF-κB signaling induced by TLR engagement, drives the translocation of p50/RelB dimers, which enables optimal DC maturation [72]. Collectively, this highlights the overlaps between canonical and alternative subunits [73]. In contrast with these immune-stimulatory properties, NF-κB activation in DCs is also critical to maintain immune tolerance. This is clearly illustrated by the spontaneous autoimmune syndrome observed in mice with conditional ablation of *Ikbkb* in CD11c^+^ DCs, which is due to impaired homeostasis of migratory DCs and a subsequent defect of Foxp3^+^ regulatory T cells (Treg cells) in lymph nodes [74]. Moreover, the alternative pathway, through IKKα activation following CD40 stimulation, was proposed to promote the expression of the immunosuppressive molecule IDO- in particular in pDC-, thereby contributing to immune tolerance, for instance through the induction of Treg cells [75,76].

These diverse functions have obvious implications in cancer immunity. Indeed, NF-κB activation is often actively dampened in DCs of the TME, which leads to reduced DC maturation and function, through mechanisms that may involve soluble factors such as Fibrinogen-like protein 2 (FGL2) and Vascular Endothelial Growth Factor (VEGF) or immune checkpoints such as PD-1 [77]. The cell-autonomous functions of NF-κB in DCs are illustrated in different settings of cancer therapy. For instance, DC-restricted ablation of *Rela* abolishes the therapeutic effect of ionizing radiation (IR) in several models of transplanted tumors, while ablation of RelB improves this therapeutic efficacy, highlighting opposing functions of canonical and alternative NF-κB subunits in this IR-induced signaling cascade [78]. In line with this, adenovirus-mediated overexpression of the RelA RHD in bone-marrow-derived DCs (BMDCs) increases their APC function and cytokine expression. Indeed, transfer of such transgenic cells to tumor-bearing mice leads to reduced tumor growth and enhanced survival [79]. In vivo administration of CD40 agonists as well as Second Mitochondria-derived Activator of Caspases (SMAC) mimetics, drives the activation of the alternative NF-κB pathway and increases the secretion of IL-12 [80]. When combined to anti-PD-1 therapy, this regimen leads to tumor rejection in a NIK-dependent fashion [81]. Finally, it is likely that many TLR agonists with therapeutic activity in cancer, act in part by enhancing NF-κB activation in DCs, though this has so far not been directly demonstrated [82,83]. Thus, NF-κB in DCs appears as an interesting target in cancer therapy. However, the roles of each subunit may vary depending on the context and deciphering their DC-intrinsic selective contributions to different cancer types and therapies may provide valuable insight and rationale into the use of DC-based immunotherapies.

#### 3.2.2. NK Cells

Through their potent cytokine production and cytotoxic functions, Natural Killers (NK) cells are powerful antitumor agents. Their enrichment in the tumor bed is associated with good prognosis in many cancers. In mouse models, NK cells are able to delay tumor growth or prevent metastasis, depending on the model (reviewed in [84]). NK cell activation and cytotoxicity largely relies on the engagement of several activating receptors. Interestingly, many of these, such as CD244, NKG2D, CD226, NK1.1, Ly49D, or Nkp30, trigger the activation of canonical NF-κB subunits in a CBM complex-dependent manner [85,86,87,88]. Similarly, IL-18, one of the master cytokines enabling NK cells function in cancer, activates NF-κB through MyD88 [89,90,91].

Consequently, NF-κB was shown to play a role in different aspects of NK cell biology. Inhibition of NF-κB with a non-degradable IκBα in NK cells reduced their proliferation [92]. Conversely, constitutive IKKβ activation leads to hyper-activated NK cells [93]. Interestingly, *Nfkb1*^−/−^ NK cells display a strong increase in proliferation, while *Rel* ablation has no effect, highlighting NF-κB1 as a negative regulator of NK proliferation [94]. In patients with *IKBKB* or *IKBKG* mutations, impaired IFNγ expression by NK cells was detected [95,96]. Similar observations were made upon chemical or genetic inhibition of canonical NF-κB in murine NK cells [86,87,92]. However, the exact contribution of each subunit remains unknown to date, as NF-κB1 and c-Rel appear to be expendable for NK cell function in vivo [94]. Finally, several reports using mouse NK cells suggested a role for canonical NF-κB subunits in the expression of *Prf1* (encoding perforin) and *Gzmb* (granzyme B) [97,98].

Despite this body of evidence, only few studies have directly investigated the function of NF-κB in NK cells in the context of cancer. It was proposed that HIF-1a expression, induced by hypoxia in the TME, inhibited IL-18-mediated NF-κB activation in tumor-infiltrating NK cells, contributing to their defective function and uncontrolled tumor progression in the RMA/S lymphoma model that exhibit impaired expression of Major Histocompatibility Complex (MHC)-I [99]. In a model of pancreatic cancer, stimulation of NK cells with the polysaccharide SEP enhanced their cytotoxicity through a TLR4/ Mitogen-Activated Protein Kinase (MAPK)/MAPKs/NF-κB pathway, which was associated with delayed tumor growth [100]. Moreover, it was proposed that the therapeutic effect of paclitaxel in breast cancer, may rely on enhanced NK cell function in an NF-κB-dependent manner [101]. Finally, CAR-NK cells engineered to express an artificial Myd88/CD40 signaling node display enhanced anti-tumor activity, likely through increased NF-κB activation [102]. Importantly, NF-κB can also exhibit a deleterious facet, as it was suggested to promote the expression of Tim-3 and NK cell dysfunction in human esophageal cancer [103]. Hence, although NF-κB mainly displays anti-tumor properties through its function in NK cells, further studies are warranted to fully apprehend their therapeutic potential and discriminate the dual roles of NF-κB in NK cell function and exhaustion.

## 4. NF-κB in Adaptive Immunity to Cancer

### 4.1. B Cells

B cells are often found in large quantities in the TME [104] but seem to have divergent functions in cancer immunity depending on the tumor context. B cells were originally shown to promote cancer growth in mouse models, for instance through their expression of soluble factors such as IL-8 or lymphotoxin (LT)-β [105,106,107], or more directly through antibody secretion [108]. In addition, a subset of “regulatory” B cells (Breg cells) can express inhibitory cytokines such as IL-10, TGF-β or IL-35 and favor tumor progression [104,109,110,111]. In line with these pre-clinical observations, the depletion of B cells using anti-CD20 mAbs, had beneficial effects in patients with colon cancer, though results were disparate in melanoma patients [112,113]. Conversely, B cells were also shown to exert anti-tumor functions through their Ab secretion (driving ADCC, complement-dependent lysis or opsonization of tumor antigens by APCs) as well as their APC capacity [114,115,116]. B-cell infiltration is associated with good prognosis in the vast majority of cancers [117]. The organization of B cells and T cells in Tertiary Lymphoid Structures (TLS) is a powerful predictor of the efficacy of PD-1-blockade therapy in human melanoma, renal cell carcinoma and sarcoma [118,119,120].

NF-κB plays important functions in the maturation and thus the function of B cells. NF-κB activation can be driven by the engagement of the antigen receptor (BCR) as well as many members of the TNFRSF family (in particular BAFF-R and CD40 for B cells). Conditional ablation of both *Rel* and *Rela* in mature B cells decreases their numbers in the spleen [121]. Similar observations were made with conditional *Ikbkb* KO animals [122]. Upon immunization with T cell-dependent antigen sheep red blood cells, c-Rel is essential for germinal center B cell maintenance, whereas RelA is required for plasma cell development [123]. Interestingly, the alternative pathway is also involved in germinal center B cell development, as shown in mice harboring conditional ablation of *Relb* and *Nfkb2* in mature B cells [124]. Interestingly, c-Rel and RelB may have complementary functions in B-cell homeostasis [125].

Despite this large body of evidence characterizing NF-κB in non-pathological states, only few studies have directly assessed the functions of NF-κB in B cells in cancer. In a mouse model of prostate cancer, it was demonstrated that IKKβ drives the expression of LT-β by B cells, which in turn augments tumor growth [107], highlighting the deleterious function of B cells in this setting. Conversely, it was recently shown that human melanoma secretomes drive the expression of many NF-κB-dependent genes in B cells, including activation markers and co-stimulation molecules such as CD69, CD30 or CD137, as well as chemokines such as CCL3 or CCL4 [112]. This enrichment was correlated with improved survival of patients and better response to anti-PD-1 checkpoint blockade therapy. However, it remains unclear whether NF-κB actually plays an active role in the orchestration of anti-tumor immune responses. More mechanistic studies, in particular in conditional KO mouse models, are therefore required to elucidate the implication of NF-κB subunits in B cells during cancer.

### 4.2. T Cells

T cells are probably the most scrutinized cells in cancer immunity. Whereas CD8^+^ T cells and IFNγ-producing Thelper (Th) 1 cells exhibit potent anti-tumor functions and are generally of good prognosis in solid tumors [126,127], their Foxp3^+^ regulatory T cell (Treg cells) counterparts are master inhibitors of cancer immunity (reviewed in [128]). Their enrichment in the TME is associated with poor prognosis in a large number of cancers [129]. Strikingly, NF-κB plays many roles in these distinct T-cell subsets.

#### 4.2.1. Effector CD4^+^ and CD8^+^ T-Cell Subsets

NF-κB was originally described as an important regulator of T-cell homeostasis, i.e., proliferation, survival and expression of specific cytokines [130]. T cells isolated from *Nfkb1*^−/−^ or *Rel*^−/−^ animals, exhibit a reduced proliferative capacity in vitro when compared to control cells [131,132]. Similarly, Shifrut et al. identified RelA as a critical regulator of human CD8^+^ T-cell proliferation using a Clustered Regularly Interspaced Short Palindromic Repeats (CRISPR)-Cas9 screening approach [133]. This observation was further validated in mice [134]. This phenotype seems to rely on both cell-intrinsic defects (e.g., impaired expression of the cell cycle factor c-Myc and the anti-apoptotic protein Bcl-xL [135,136]) and extrinsic factors, in particular the reduced production of IL-2, which when added to cultures rescues the proliferative defects of *Rel*-deficient T cells [135]. In addition, NF-κB has cell-autonomous roles in the polarization towards different flavors of Th cells. RelA and c-Rel are crucial for the expression of RAR-related orphan receptor (ROR)γt in differentiating Th17 cells and have been involved in the expression of IFNγ by Th1 cells [137,138,139,140]. A recent study pinpointed possible functions of RelA and NF-κB2 in the expression of *Gata3* and *Il4*, respectively, by murine Th2 cells [141]. Finally, Th9 cells, which can exhibit potent anti-tumor properties, may rely on either canonical or alternative NF-κB activation, depending on the stimulus [142,143]. NF-κB is also involved in the maturation and function of CD8^+^ T cells. For instance, RelA activation downstream of TCR engagement, is required for optimal expression of *Eomes* [144]. The expression of *IFNγ* by CD8^+^ T cells is also NF-κB dependent, as shown in mice with conditional ablation of *Rela* in T cells [139], and in patients carrying loss-of-function mutations in *IKBKB* and *REL* [96,145]. Finally, PKC-θ and Bcl10 are required for the cytotoxic function of CD8^+^ T cells in vitro, implying of the need for canonical NF-κB activation in this mechanism [146].

Altogether, this suggests that NF-κB likely plays a role in controlling T-cell activation, survival and/or function in the context of cancer. In line with this, NF-κB activation is dysregulated in T cells from patients and mice with renal cancer [147,148]. Moreover, peripheral anergic T cells, which resemble tumor-infiltrating exhausted T cells, exhibit reduced RelA nuclear translocation [149]. More directly, the function of NF-κB in T-cell immunity to cancer was explored in a seminal study by Barnes and al, where the authors used mice carrying conditional ablation of *Ikbkb* in total T cells. Inhibition of canonical NF-κB activation leads to increased tumor growth in an immunogenic model of fibrosarcoma [150]. This was associated with reduced effector function and cytotoxic activity of CD8^+^ T cells. This phenotype was also observed in *Carma1*-deficient mice. In a reverse set of experiments, the same group found that constitutive activation of IKKβ (Ikkβ-CA) in T cells, strongly decreases tumor growth in a CD8^+^ T cell-dependent manner [151]. Engineering WT or TCR-transgenic CD8^+^ T cells with an active form of IKKβ (Ikkβ-CA), enables potent tumor rejection. Similarly, ablation of *Tnfaip3* in TCR-Tg T cells, which increases canonical NF-κB activation, augments their therapeutic potential in transplanted mouse melanoma [152]. Overall, this highlights an interest for genetically-augmented NF-κB activation in T-cell-based cancer therapies. It would now be interesting to decipher which cell types and which NF-κB subunit(s) are required for this control of tumor growth. A recent report suggests that p50/p50 homodimers are able to directly repress *Gzmb* expression in CD8^+^ T cells, consequently, *Nfkb1*^−/−^ mice exhibit a partial resistance to colon tumorigenesis [153]. In addition, as described above, NF-κB subunits orient the polarization toward Th cell subsets. Importantly, both protective and deleterious functions were attributed to discrete Th cell subsets in cancer immunity [154]. It is thus likely that this function will impact tumor growth, though the link has not yet been determined. Finally, NF-κB seems to be important for the reinvigoration of T cells following checkpoint-blockade therapies, and this point will be largely discussed below in Section 5.1.

#### 4.2.2. Foxp3^+^ Treg Cells

Unlike its function in promoting immunity by controlling effector T cell function, NF-κB is also a master regulator of Treg cell biology and thus regulates immune tolerance. Activation of RelA and c-Rel, through a PKCθ/CBM/IKK axis, is critical for the initiation of Foxp3 expression in immature Treg cells in the thymus [155,156,157,158]. Consequently, germline or conditional ablation of one of the aforementioned molecules leads to a significant decrease in the number of Foxp3^+^ Treg cells in the thymus and secondary lymphoid tissues. In addition, NF-κB activation is required to maintain the homeostasis and identity of mature peripheral Treg cells. Treg-restricted ablation of *Carma1*, *Bcl10*, *Malt1*, *Ikbkb* or *Rela* using the Foxp3-CRE strain, leads to a multifocal and lethal autoimmune syndrome, characterized by the loss of Treg signature genes and the expression of effector genes such as inflammatory cytokines [158,159,160,161,162]. On the other hand, activation of the alternative pathway seems to have a rather deleterious effect, as transgenic expression of *Nik* [163], as well as ablation of *Nfkb2* [164,165], also drive autoimmune symptoms. The latter can be rescued by additional ablation of *Relb*, which by itself is expendable for Treg cell homeostasis [166].

This master function of NF-κB in Treg cells has strong implications in cancer immunity. Indeed, the growth of transplanted tumors is strongly reduced in mice with Treg-specific deletion of either *Carma1*, *Bcl10* or *Malt1* [159,160,161]. This is consistent with our original observation that melanoma burden is significantly impaired in mice with conditional *Rel* ablation in Treg cells [167]. Interestingly, germline ablation of *Rel* gives rise to the same phenotype [48]. Mechanistically, ablation of the CBM/c-Rel axis leads to the generation of ‘unstable’ Treg cells that produce important amounts of IFNγ and TNF. Thus, Treg cells become potent anti-tumor cells, as the transfer of *Carma1*^−/−^ Treg cells is sufficient to reduce cancer growth in an IFNγ-dependent fashion [159]. In line with all these observations, administration of Malt1 inhibitors (mepazine, MI-2), IKKβ inhibitor (KINK-1) or c-Rel inhibitors (pentoxifylline, IT-603 or R96A) reduces Treg cell ‘fitness’ in the tumor, i.e., their accumulation and/or their function [48,159,162,167,168]. Consequently, these drugs have a significant therapeutic effect in different pre-clinical models of cancer, when administered alone or combined with anti-PD-1/PD-L1 mAbs, tumor vaccines or chemotherapy.

Collectively, this demonstrates that NF-κB is at the interface between T-cell immunity and tolerance to tumors. This highlights a strong rationale for targeting this pathway in cancer immunotherapy.

## 5. NF-κB Modulation at the Era of Immunotherapy: Activation or Inhibition?

### 5.1. NF-κB Activation to Enhance Effector T-Cell Function in Cancer

#### 5.1.1. Putative Functions of NF-κB in T-Cell-Targeting Immunotherapies

Cancer immunotherapies developed in the past decades have revolutionized patient care. In particular, immune checkpoint-blockade therapies such as anti-CTLA-4 and anti-PD-1/PD-L1 antibodies, and CAR-T-cells, appear as game changers for the treatment of numerous solid cancers and hematological malignancies, respectively. However, these agents are not effective in all patients, and it is therefore critical to understand the molecular mechanisms underlying the response and resistance to such therapies. Interestingly, a number of studies converge to consider the NF-κB pathway as a possible mediator of the anti-tumor properties of current immunotherapies.

Bulk and single-cell transcriptomic analyses of tumor specimens obtained from patients before and after anti-CTLA4 and/or anti-PD1 administration, have unequivocally demonstrated that the up-regulation of NF-κB-dependent genes in both tumor and immune cells, underlies the clinical response to checkpoint-blockade therapies [169,170]. In particular, a number of supposedly NF-κB-regulated cytokines and chemokines are up- and down-regulated in patients who respond to immunotherapy [171]. However, it remains unclear whether NF-κB activation is a proxy of T-cell activation or is actually required for improving anti-tumor responses. Mechanistically, it was proposed that the engagement of either CTLA-4 or PD-1 on T cells, may reduce NF-κB activation through the recruitment of Src homology region 2 domain-containing phosphatase-2 (SHP-2) and Protein Phosphatase (PP)2A, leading to reduced signaling downstream of TCR and CD28 [172,173,174,175]. Subsequently, CTLA-4 or PD-1-blockade would lead to increased NF-κB activity in T cells. In line with this hypothesis, it was shown that CD28 engagement, which drives NF-κB activation among other pathways, is critical for the efficacy of anti-PD-1 therapy in mice [176]. Moreover, the expression of IFNγ and CD127, two canonical NF-κB-dependent genes, are required for tumor clearance upon CTLA-4/PD-1 dual blockade [177].

Improvement of NF-κB activity in cellular therapies to cancer is also a promising approach. As stated above, ablation of *Tnfaip3* (encoding the NF-κB inhibitor A20) or expression of Ikkβ-CA augments the anti-tumor activity of transferred T cells [151,152]. Recently, a genome-wide CRISPR approach revealed that ablation of the RNA helicase Dhx37 in murine T cells improved tumor clearance by increasing RelA activity [178]. In addition, possible implication of NF-κB was reported in the expansion of CD19- CAR-T-cell carrying a 4-1BB signaling domain. Indeed, engagement of the CAR in these cells drives sustained NF-κB activation in a TRAF1-dependent mechanism [179]. Both the canonical and alternative NF-κB pathway are activated following CD19 recognition, and expression of a dominant-negative NIK peptide reduces CAR-T cell expansion and survival ex vivo [180]. Further studies are now required to elucidate the direct function of discrete NF-κB subunits in the therapeutic effect of CAR-T cell administration in vivo.

These findings strongly advocate for a beneficial role of NF-κB activation in the response to checkpoint-blockade and CAR-T-cell-based therapies, though direct evidence of its requirement is lacking.

#### 5.1.2. Stimulation of NF-κB by TLRs and TNFRSFs Agonistic Reagents as Novel Immunotherapies

The therapeutic potential of TLR agonists in mouse models of cancer has been largely documented (reviewed in [83]). This has led to numerous clinical trials in which patients are treated with synthetic TLR3, 4, 7 and 9 agonists in combination with other immunotherapies or conventional treatments [181]. Because TLR signaling through NF-κB is a master inducer of innate immune cell activation, maturation and inflammatory cytokine expression, it is likely that the anti-tumor effect of these drugs relies on NF-κB activation. Similar hypotheses can be made for Stimulator of Interferon Genes (STING) and CD40 agonists, which drive NF-κB activation in DCs (and B cells for CD40) [78,182,183]. In addition, a number of initiatives currently aim at exploring the therapeutic potential of antibodies triggering activation of “stimulatory” checkpoints on T cells, i.e., co-stimulation receptors of the TNFRSF family such as 4-1BB or Glucocorticoid-Induced TNFR (GITR). Engagement of these receptors has been proposed to induce or sustain NF-κB activation in combination with TCR stimulation [184]. Engagement of all these receptors improves immunity to cancer and reduces tumor growth in pre-clinical models [185]. These observations are contributing to pushing these agents toward clinical applications [185]. However mechanistically, it is still unclear whether the therapeutic efficacy of these molecules relies on NF-κB activation. Furthermore, the specific delineation of NF-κB subunits in this process is unknown. Novel tools, such as mice carrying lineage-specific NF-κB ablation, will help unravel these issues and may provide a mechanistic rationale for the use of these new therapies.

### 5.2. NF-κB Inhibitors as Immunotherapeutic Agents?

#### 5.2.1. Modulation of Tumor Immunity by NF-κB Inhibitors

At odds with this possible protective function in T cells, it is clear that NF-κB can have pathogenic functions in cancer, in particular through its well-established roles in malignant cells. Consequently, a myriad of inhibitors of the NF-κB pathway have been tested for their anti-cancer properties, with encouraging results in pre-clinical models. These compounds were shown to directly impact tumor cell survival, proliferation, or invasiveness in in vitro assays and in vivo. In addition, it is becoming evident that NF-κB inhibitors largely modulate the tumor immune response. Indeed, therapeutic intervention at each step of the NF-κB pathway (i.e., Cellular Inhibitor of Apoptosis Protein (cIAPs), CBM and IKK complexes, IκBs, and c-Rel) was shown to enhance cancer immunity to some extent. A large panel of modifications in the immune composition of the tumor infiltrate and the function of immune cells was described, and some relevant examples are highlighted in Table 1. The precise reasons for this immune stimulation are not clearly defined, though the following mechanisms can be proposed (Figure 2):

(i) One possible mechanism relies on tumor cell-targeting effects of NF-κB inhibitors, which would in turn enhance immune responses. Tumor cell death induced by Bortezomib was proposed to enhance DC maturation and stimulation of potent anti-tumor effector T cells and NK cells [197]. Moreover, the chemokine expression profile of tumor cells may be affected by NF-κB inhibition, leading to enhanced tumor infiltration, as shown with the IKK inhibitor BMS-345541 [198] or curcumin [199]. NF-κB is also well-known to favor the expression and stability of PD-L1 at the surface of tumor cells, with the administration of curcumin to breast-tumor bearing mice decreasing PD-L1 expression and subsequently enhancing CD8^+^ T-cell infiltration [14].

(ii) Aside from these tumor-targeting effects, NF-κB inhibitors likely function by relieving the suppression exerted by MDSCs and Treg cells. The latter, in particular, are impaired in the tumor tissue following administration of mepazine, KINK-1 and pentoxifylline (PTXF) (reviewed in [200]). Similarly to what was highlighted in Section 4.2.2 using genetically-engineered mice, agents targeting the NF-κB pathway are able to specifically decrease Treg cell numbers and impair their stability, leading to the expression of effector cytokines such as IFNγ or TNF, in the TME but not in lymphoid tissue. This suggests that these therapies should cause limited adverse effects in normal tissues. Thus, the effects of NF-κB inhibitors on cancer immunity vary depending on the drug and cancer model. Generally, all of these reagents lead to an increase in the number and/or the function of cytotoxic CD8^+^ T cells in the tumor microenvironment. Importantly, these changes in immune cells do not seem to be solely a side effect of tumor cell targeting, but rather play an active role in cancer elimination. For instance, administration of mepazine or PTXF fails to reduce tumor burden in *RAG1*^−/−^ animals that are devoid of B and T cells [159,167].; In addition, the therapeutic effect is lost when CD8^+^ T cells are depleted by a monoclonal antibody, highlighting the importance of T cells in the therapeutic potential of NF-κB inhibitors.

#### 5.2.2. Combination of NF-κB Inhibitors and Other Immunotherapies

The main challenge in cancer immunotherapy is now to identify drugs that are able to overcome innate or acquired resistance to anti-PD-1 or anti-CTLA-4 checkpoint-blockade therapies. Combination treatments associating NF-κB inhibitors to other immunotherapies are emerging as a promising step forward in cancer care (Table 2). In the context of anti-PD-1-therapy-resistant tumors, the co-administration of either curcumin, bortezomib, SMAC mimetics, mepazine, PTXF or R96A enhances anti-tumor immunity and delays cancer growth. The combination of anti-PD-1 and bortezomib is currently investigated in clinical trials in multiple myeloma patients. NF-κB inhibitors can also be associated with cytokines (IL-12, IL-18, IFNα), oncolytic viruses or other antibody-based therapies (anti-CTLA4, anti-CD38). The relative lack of efficacy of NF-κB inhibitors observed in clinical trials of the past decades may therefore be overcome by their association with immunotherapies.

### 5.3. Limitations of NF-κB-Targeting Therapies and Future Directions

Despite great expectations following the early development of NF-κB inhibitors for the treatment of inflammatory diseases and cancer, very few of these molecules have reached advanced phases of clinical trials. This is due to either modest efficacy or severe adverse effects. In light of recent progress in understanding NF-κB mediated regulation of immune responses, one could now envision a revival of this class of drugs. However, several aspects of the immense power of NF-κB must be taken into account before we can safely move towards large-scale clinical applications:

*The type of cancer:* It is very likely that the putative efficacy of NF-κB inhibitors will depend on the type of malignant tissue and its immune environment. It was proposed that NF-κB inhibition would be especially relevant for the treatment of inflammation-driven cancers and NF-κB-“addicted” tumors [213]. Hypothetically, NF-κB activation may in some settings have deleterious roles in tumor-cell autonomous biology, e.g., apoptosis or chemokine/cytokine expression, though there is to our knowledge, no direct evidence of this. Moreover, the composition and quality of the immune response must be carefully evaluated when it comes to NF-κB inhibition. In the case of cold tumors, or conversely in cancers undergoing a partial anti-tumor response, it is possible that NF-κB inhibition may have no effect or even reduce the activity of effector cells.

*The cell-specific targeting:* In pre-clinical models of cancer, it is likely that NF-κB inhibitors are partially efficient because they primarily target suppressive cells that are largely enriched in the tumor environment. However, it is important to note that NF-κB is also a master regulator of potent anti-tumor effector cells as described above. Therefore, the duration, dosage and specificity of these treatments must be tightly controlled to avoid deleterious effects. For instance, Heuser et al. showed that administration of high doses of KINK-1 impaired CD8 T-cell function, whereas lower doses displayed therapeutic effects in melanoma [162]. Recent advances in drug engineering may also help in the development of more specific, thus more potent and safer, drugs. Antibody-drug conjugates, allowing the delivery of the payload to defined cell types, are currently under development [214]. This system could be used to deliver NF-κB inhibitors to either tumors cells or immunosuppressive cells. Consistently, curcumin conjugated to anticancer Abs, displays promising tumoricidal activity in pre-clinical models of melanoma and glioblastoma [215,216].

*The precise delineation of the roles of NF-κB subunits:* It is increasingly evident that NF-κB subunits exhibit distinct functions in immune cell subsets and cancer cells. These selective contributions may underlie the lack of efficacy of certain NF-κB-targeting agents and explain their toxicity. The use of conditional knock-out and knock-in animal models, as well as large-scale CRISPR screens and modern sequencing techniques, should provide, both at the whole-organism and single cell levels, the gene expression patterns controlled by each member of the NF-κB family. In addition, the modeling of interactions between tumor cells and their microenvironment using organoid systems, will allow the development of screening approaches and the identification of selective modulators of the NF-kB pathway. This will undoubtedly help the discovery of unique therapeutic targets for the treatment of cancer.

## 6. Conclusions

Immune responses to cancer are complex and vary depending on the type of malignancy. However, the growing knowledge on the dozens of innate and adaptive immune cell subsets has enabled the identification of novel therapeutic targets. In this review we have highlighted the multifaceted roles of 5 members of the NF-κB family of transcription factors, in orchestrating tumor immunity. Targeting NF-κB in cancer may not be a revolutionary concept as it finds its roots in the early 90s, but there is now a strong rationale for the revival of NF-κB-targeting therapies. Indeed, in addition to their direct effect on tumor cell survival, proliferation or metastatic potential, modulators of the NF-κB pathways are also likely to act as immune-stimulatory agents. New efforts must now be directed at identifying agents able to selectively activate or repress specific NF-κB subunits, and to ensure their efficacy, delivery mode and safety.

## Figures and Tables

**Figure 1 cells-10-00355-f001:**
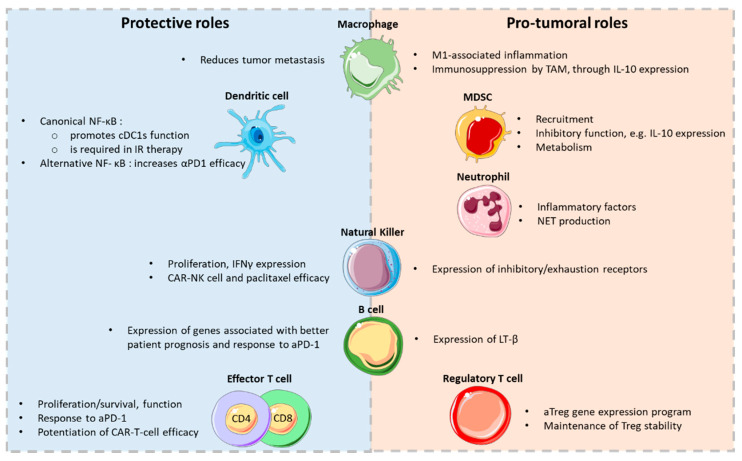
Dual roles of NF-kB in tumor immunity. The figure depicts some of the known functions of NF-kB in immune cell subsets in the tumor environment. Some figure elements were obtained from Servier Medical Art (https://smart.servier.com/).

**Figure 2 cells-10-00355-f002:**
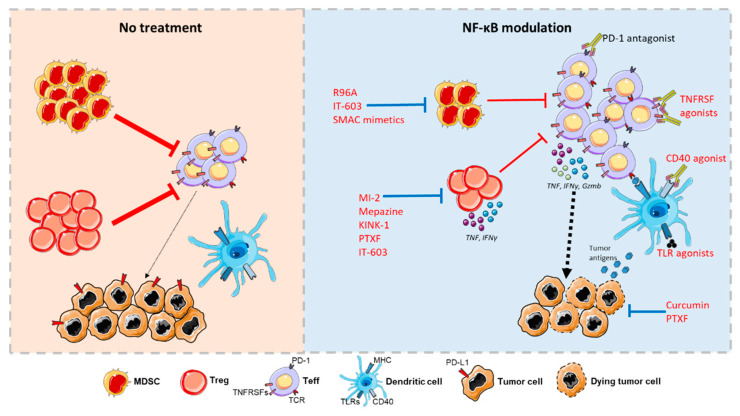
Proposed mechanism for the immunotherapeutic effects of NF-κB modulating agents. In the absence of treatment, MDSCs and Treg cells actively inhibit CD4^+^ and CD8^+^ Teff cells; tumor-derived Ag presentation by DCs is low. NF-κB inhibitors increase the numbers and activity of Teff cells by decreasing the recruitment and fitness of MDSCs and impairing Treg cell identity, relieving the suppression of Teff cells. Moreover, these treatments reduce PD-L1 expression and induce tumor cell death, leading to enhanced Ag-presentation by DCs. Conversely, stimulation of TNFRSFs and TLRs on DCs and Teff cells activates NF-κB to increase their maturation numbers and function. All these agents potentiate the therapeutic effect of anti-PD-1 mAbs. Some figure elements were obtained from Servier Medical Art (https://smart.servier.com/).

**Table 1 cells-10-00355-t001:** Effect of NF-κB inhibitors on cancer immunity. This table highlights some of the studies aiming at evaluating monotherapy-induced immune responses in the context of cancer in vivo. In vitro assays and other diseases are not discussed.

Agent/Mechanism	Tumor Model	Impact on Immune Cells	References
Mepazine/MI-2, MALT-1 inhibitor	Mouse melanoma	↑ CD8^+^ T cells, ↑ IFNγ by Treg cells in TILs; efficacy lost in RAG^−/−^ mice	[159]
Curcumin, prevents IκBα degradation, NF-κB translocation	Mouse breast, colon, lung cancer	↑ T cells and IFNγ expression, ↓ Treg cell proportion in TILs	[186,187,188]
	Mouse breast, bladder lung cancer	↓ MDSCs in TILs, ↑M1 macrophages	[189,190,191]
	Mouse glioblastoma	↑ NK cells; NK-cell-dependent therapeutic effect	[192]
	Patients with lung cancer	↑ IFNγ expression by CD4^+^ T cells, ↓ Treg cell proportions in PBMCs	[193]
KINK-1, IKKβ inhibitor	Mouse melanoma	↑ CD8^+^ T-cell infiltration and function in tumors	[162]
SMAC mimetics (CIAP antagonists), inhibit canonical NF-κB, enhance alternative NF-κB	Mouse Multiple myeloma	↑macrophage phagocytosis; macrophage-dependent therapeutic effect	[194]
	Mouse glioblastoma, breast cancer	↑ T-cell in TILs; CD8^+^ T-cell-dependent therapeutic effect	[195]
IT-603, c-Rel inhibitor	Mouse thymoma, melanoma	↓ circulating T cells; ↑ IFNγ expression by Teff/Treg cells in TILs	[196,167]
PTXF, c-Rel inhibitor	Mouse breast cancer, melanoma	↑ CD8^+^ T cells, ↓ Treg cells in TILs; efficacy lost in RAG^−/−^ mice	[168,167]
R96A, c-Rel inhibitor	Mouse melanoma	↓ inhibitory function of MDSCs	[48]

**Table 2 cells-10-00355-t002:** Examples of combination of NF-kB inhibitors with other immunotherapies.

Organism.	Combination Therapy	Cancer Type(s)	References
Mouse Studies	RELA shRNA + αPD-1	Hepatocellular carcinoma	[201]
	curcumin + αCTLA-4	Breast cancer	[14]
	curcumin + αPD-1/PDL-1	Diverse	[189,199,202]
	Bortezomib + IL12	Multiple myeloma	[203]
	Bortezomib + reovirus	Multiple myeloma	[204]
	Mepazine + αPD-1/PD-L1	Melanoma	[159,160]
	BAY11-7082 + rmIL-18	PDAC	[205]
	SMAC mimetics + oncolytic VSV	Breast cancer	[206]
	SMAC mimetics + αPD-1	Diverse	[81,207]
	PTXF/IT-603/R96A + αPD-1/PD-L1	Diverse	[48,167]
	PTXF + αPD-1 + chemotherapy	Breast cancer	[168]
Human Studies	Bortezomib + αCD38 + dexamethasone	Multiple myeloma	[208]
	Bortezomib + αPD-1	Multiple myeloma	[209]
	Bortezomib + αSLAMF7	Multiple myeloma	[210]
	Bortezomib + Interferon α2B	Melanoma	[211]
	SMAC mimetics + αPD-1	Diverse	[212]

## Data Availability

No new data were created or analyzed in this study. Data sharing is not applicable to this article.

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
