# Peer review of "NF-κB in Cancer Immunity: Friend or Foe?"

_cells, 2021, doi:10.3390/cells10020355_

Round 1

Reviewer 1 Report

The paper by Lalle et al. “NF-kB in cancer immunity: friend or foe?” the authors give an interesting overview of the recent literature of NF-kB effects on immune cells and immune response to cancer. The article is coherent and well written and with exception of point 1 only minor corrections have to be made.

  1. Unfortunately, in the part 5.2.1 a formatting error occurred which makes it difficult to read and understand part 5.2.1. From line 513 the normal text is formatted as explanatory caption of figure 2. Part (i) suddenly stops in the middle of the sentence (line 517). Also the sentence in line 526 is not conclusive and the caption of figure 2 seems to contain parts of the main text (ii). This should be carefully revised.

  1. There are two email addresses of the corresponding author, the first is not complete.
  2. The two pathways of NF-kB activation are described in part 2. While the “canonical” pathway is highlighted by italian printing, the “alternative” is not (line 65).
  3. The indication of (ref. 157, 158) makes no sense to the reader in line 167.
  4. The authors should reach an agreement about a uniform spelling (e.g. CD8-T-cell vs. CD8+ T cell (line 381, 382)).
  5. The authors should check the usage of italic printing for genes and normal for proteins (e.g. Rel: line 414 /line 415).
  6. All abbreviations should be defined in parentheses the first time they appear in the text and used consistently thereafter (e.g. NET production (line 194), Tconv (line 211), conventional dendritic cell type 1 (cDC1s) (line 206)…..).
  7. In line 281 to 285 a NF-kB activation in tumor infiltrating NK cells with an impact on tumor progression is described. Which kind of tumor model was used in the study?
  8. A legend to table 1 should be added.
  9. My last point is more an expression of interest than a point if criticism: The authors report about the obstacles in the research of cancer cell TME interactions and consider animal models and CRISPR screens as important techniques to discover new therapeutic targets and treatment options. In recent years, personalized medicine and in this context the research status of organoid models has made progress. Meanwhile, organoids are also used for recapitulating the immune TME and investigations of cancer immunobiology. Are there any publications yet, which deal with organoids, immunobiology and NF-kB? If this is the case, it would be great to discuss this in the article. 

Author Response

We thank the reviewer for his/her careful assessment of our manuscript. We have addressed the reviewer’s comments below and in the edited version of our paper.

1. Unfortunately, in the part 5.2.1 a formatting error occurred which makes it difficult to read and understand part 5.2.1. From line 513 the normal text is formatted as explanatory caption of figure 2. Part (i) suddenly stops in the middle of the sentence (line 517). Also the sentence in line 526 is not conclusive and the caption of figure 2 seems to contain parts of the main text (ii). This should be carefully revised. 

We apologize for this formatting error. It has been revised in the edited version of the manuscript.

2. There are two email addresses of the corresponding author, the first is not complete.

The incomplete e-mail address has been removed.

3. The two pathways of NF-kB activation are described in part 2. While the “canonical” pathway is highlighted by italian printing, the “alternative” is not (line 65).

We have edited the text accordingly.

4. The indication of (ref. 157, 158) makes no sense to the reader in line 167.

We have removed these unformatted references.

5. The authors should reach an agreement about a uniform spelling (e.g. CD8-T-cell vs. CD8+ T cell (line 381, 382)).

We have edited line 381 to ‘CD8+ T cell’.

6. The authors should check the usage of italic printing for genes and normal for proteins (e.g. Rel: line 414 /line 415).

We have carefully checked the printing of each gene/protein in the edited manuscript.

7. All abbreviations should be defined in parentheses the first time they appear in the text and used consistently thereafter (e.g. NET production (line 194), Tconv (line 211), conventional dendritic cell type 1 (cDC1s) (line 206)…..).

We have carefully checked and edited the manuscript accordingly.

8. In line 281 to 285 a NF-kB activation in tumor infiltrating NK cells with an impact on tumor progression is described. Which kind of tumor model was used in the study?

In this study (ref 99), the authors used the RMA-S lymphoma cell line. These cells have reduced expression of MHCI/peptide complexes, which renders them sensitive to NK-cell-mediated cytotoxicity (see De Bruijn, EJI 1991 doi: 10.1002/eji.1830211210). We have edited the manuscript to specify the cell type.

9. A legend to table 1 should be added.

The ‘legend’ for each table now stands next to its title.

10. My last point is more an expression of interest than a point if criticism: The authors report about the obstacles in the research of cancer cell TME interactions and consider animal models and CRISPR screens as important techniques to discover new therapeutic targets and treatment options. In recent years, personalized medicine and in this context the research status of organoid models has made progress. Meanwhile, organoids are also used for recapitulating the immune TME and investigations of cancer immunobiology. Are there any publications yet, which deal with organoids, immunobiology and NF-kB? If this is the case, it would be great to discuss this in the article. 

We thank the reviewer for this excellent suggestion. We fully agree that the development of organoids/tumoroids will help understanding the relationships between cell types of the tumor microenvironment. This will provide a mechanistic rationale to the use of specific NF-kB targeting drugs, and enable new strategies of drug screening, In fact, the deleterious functions of NF-kB as a driver of tumorigenesis were clearly recapitulated in organoid models. In such systems, it was also demonstrated that NF-kB activation in tumor cells (for instance, driven by TLR stimulation or inflammatory cytokines) impacts the expression of cytokine and chemokine expression, thereby affecting the recruitment of immune cells such as neutrophils. However, after a broad search in the literature, we found no direct assessment of the effect of immune cell-intrinsic NF-kB inhibition/activation (using genetic ablations or chemical inhibitors) on immune responses to cancer organoids. A single paper measured the impact of WT and RelaKO BMBM-conditioned medium, on the formation of glioblastoma spheroids (and found no impact of NF-kB ablation) (ref 28 in our manuscript).

We have added a few sentences in section 5.3, to emphasize the importance of organoid models in the discovery of new NF-kB-related immunotherapeutics.

Reviewer 2 Report

This review comprehensively summarized the current state of knowledge of the role of NF-kB in the function of immune cells in the tumor environments. The summary tables are quite comprehensive and informative. Based on the literature the authors propose possible mechanisms in the modulation of tumor immunity by NF-kB inhibitors, represented in an interesting figure 2. However, the legend of the figure or the text that comments on the figure present some formatting or editing misplace of the text that needs to be solved:

Line 512 (i) One possible…. Where is (ii)? Is “(ii) Aside from…” on line 530 part of the text or is part of the figure 2 legend?

Line 517 seems to be trunked “infiltration, as”?

Lines 526-527 check the test underlined “All these agents potentiate the therapeutic effect of anti-PD-1 mAbs. Some figure 526 elements were obTable 345541.  or curcumin [199]. NF-κB is also well-known to favor the expres-527 sion and stability of PD-L1 at the surface of tumor cells, with the administration of curcumin to 528 breast-tumor bearing mice decreasing PD-L1 expression and subsequently enhancing CD8+ T-cell 529 infiltration [14].

Line 339: “2. T cells” add 4 as “4.2 T cells

Related to the content of section 4.2.1, are there any updates on NF-kB and tumor viruses such as HTLV that might be worth mentioning?

Minor point:

Line 127: it would be useful to specify as “Lys-CRE mouse strain”

Line 129 “of breast, and ovarian cancer and glioblastoma”

Line 202 “, Through” erase “,”

Author Response

We thank the reviewer for his/her positive assessment of our manuscript. We have carefully addressed the reviewer’s comments below and in the edited version of our paper.

  1. However, the legend of the figure or the text that comments on the figure present some formatting or editing misplace of the text that needs to be solved:
  2. Line 512(i) One possible…. Where is (ii)? Is “(ii) Aside from…” on line 530 part of the text or is part of the figure 2 legend?
  3. Line 517 seems to be trunked “infiltration, as”? 
  4. Lines 526-527 check the test underlined “All these agents potentiate the therapeutic effect of anti-PD-1 mAbs. Some figure 526 elements were obTable 345541.  or curcumin [199]. NF-κB is also well-known to favor the expres-527 sion and stability of PD-L1 at the surface of tumor cells, with the administration of curcumin to 528 breast-tumor bearing mice decreasing PD-L1 expression and subsequently enhancing CD8+ T-cell 529 infiltration [14].

We apologize for these formatting errors due to the insertion of the figure and its legend in the Word document. It has been revised in the edited version of the manuscript.  

  1. Line 339: “ T cells” add 4 as “4.2 T cells

We have edited the text accordingly.

  1. Related to the content of section 4.2.1, are there any updates on NF-kB and tumor viruses such as HTLV that might be worth mentioning?

We thank the reviewer for this interesting comment. Extensive literature has demonstrated modulation of NF-kB activation by oncoviruses such as HTLV-1, in T cells. In particular, Tax-mediated constitutive activation of canonical and alternative NF-kB, is believed to mediate cell transformation, proliferation and survival. In contrast the antisense protein HBZ reduces NF-kB activation. Importantly, the cytokine profile of HTLV-1-transformed T cells is dramatically impaired, with an upregulation of IL-10 expression favoring immune suppression and tumor escape. This represents an additional facet of NF-kB as a driver of oncogenesis. However, to our knowledge, it is still not clear whether NF-kB is actually responsible for the immunosuppressive status of the infected T cell, and whether this directly impairs tumor immunity. The complex hijacking of different host pathways by viruses, complicates the full dissection of molecular determinants of transformation/proliferation/cytokine expression. Consequently, we have chosen not to tackle this issue in our manuscript.

  1. Line 127: it would be useful to specify as “Lys-CRE mouse strain”

We have edited line 127 accordingly : ‘Lysozyme (LysM)-CRE strain’.

  1. To Line 129 “of breast, and ovarian cancer and glioblastoma” /Line 202 “, Through” erase “,”

We have edited the text accordingly.